# Prognostic Significance of Venous-to-Arterial CO_2_ Difference in Critically Ill Patients After Major Abdominal Surgery

**DOI:** 10.3390/biomedicines13092295

**Published:** 2025-09-18

**Authors:** Gyeo Ra Lee, Eun Young Kim

**Affiliations:** Division of Trauma and Surgical Critical Care, Department of Surgery, Seoul St. Mary’s Hospital, College of Medicine, The Catholic University of Korea, Seoul 137-701, Republic of Korea; leegyoura@naver.com

**Keywords:** abdominal surgery, micro-circulation, postoperative prognosis, ScvO_2_, venous-to-arterial carbon dioxide partial pressure difference

## Abstract

**Purpose:** The venous-to-arterial carbon dioxide partial pressure difference [P(v-a)CO_2_] reflects the adequacy of tissue perfusion, with elevated values suggesting impaired clearance of CO_2_. While its prognostic role has been investigated in septic shock and high-risk surgery, evidence in postoperative critically ill patients remains limited. This study aimed to evaluate the prognostic value of ΔP(v-a)CO_2_ after major abdominal surgery and its relationship with microcirculatory markers. **Methods**: We retrospectively analyzed 86 patients admitted to the intensive care unit (ICU) after major abdominal surgery between September 2020 and October 2023. Arterial and central venous blood gas analyses were performed immediately postoperatively and at 24 h. Patients were stratified into groups according to ΔP(v-a)CO_2_ (≤ 0 vs. >0). Postoperative outcomes and correlations with central venous oxygen saturation (ScvO_2_) were assessed. **Results**: In the subgroup analysis of patients with an initial P(v-a)CO_2_ > 6 mmHg, those in the ΔP(v-a)CO_2_ > 0 group required mechanical ventilation (54.5% vs. 22.2%, *p = 0.033*) and continuous renal replacement therapy (36.4% vs. 8.9%, *p = 0.020*) more frequently, with longer durations of both interventions (*p = 0.011* and *p = 0.016*, respectively). ICU length of stay and the incidence of acute kidney injury were significantly lower in the ΔP(v-a)CO_2_ ≤ 0 group. In addition, a modest negative correlation was observed between ScvO_2_ measured at 24 h postoperatively and ΔP(v-a)CO_2_. **Conclusions**: ΔP(v-a)CO_2_ may serve as a useful marker for postoperative risk stratification in critically ill patients undergoing major abdominal surgery. However, given the retrospective design, small sample size, and single-center setting, these findings should be considered hypothesis-generating and require confirmation in larger, prospective multicenter studies.

## 1. Introduction

The primary objective of hemodynamic resuscitation in patients admitted to the intensive care unit (ICU) post-surgery is the normalization of reduced microcirculation due to intraoperative hypoperfusion or tissue injury, to meet the oxygen and metabolic demands of major organs. Although blood flow to vital organs is typically maintained by the host’s auto-regulatory mechanisms, it becomes compromised in shock or clinically deteriorating conditions. In these cases, maintaining appropriate blood flow to major organs is challenging, thus making the accurate assessment and monitoring of blood flow status crucial. Throughout the resuscitation process, the restoration of impaired microcirculation is mainly assessed indirectly through macrohemodynamic variables such as mean arterial pressure or stroke volume, which refers to the amount of blood ejected by the left ventricle with each heartbeat and reflects the effectiveness of cardiac output. However, these variables do not adequately represent impaired microcirculation, which may result from factors such as increased blood viscosity, endothelial dysfunction, or microthrombi formation. It should also be considered that in certain cases, particularly following excessive fluid administration, hemodilution may occur, potentially offsetting the impact of viscosity-related impairment. Moreover, fluid resuscitation based on incorrect parameters or a high blood pressure target above a mean arterial pressure (MAP) of 65 mmHg could lead to an inappropriate positive fluid balance. Previous studies by Opsina et al. [1] and Pottecher et al. [2] have shown that responses to fluid administration do not directly correlate with effects on changes in systemic circulation status. Therefore, it is essential to use appropriate markers that more accurately reflect the status of microcirculation in critically ill patients, particularly in those at risk of tissue or endothelial injury due to surgical stress or sepsis-related microthrombi formation. The venous-to-arterial carbon dioxide tension difference [P(v-a)CO_2_] refers to the difference in the partial pressure of carbon dioxide (CO_2_) between venous and arterial gas analyses. P(v-a)CO_2_ is primarily an indicator of the adequacy of cardiac output relative to CO_2_ production, rather than a pure marker of microcirculation. When cardiac output is low, CO_2_ clearance decreases and venous PCO_2_ rises relative to arterial PCO_2_, a mechanism well explained by Fick’s principle for CO_2_ [3,4]. This parameter has been shown to be associated with mortality, organ dysfunction, and adverse postoperative outcomes in high-risk surgical patients. The usual pathological threshold is approximately 6 mmHg, and many studies have used values above this threshold as a sign of hypoperfusion or inadequate flow, with poor prognosis in septic shock and high-risk surgical populations [3,4,5,6]. In addition, ScvO_2_ measurement supplemented with lactate has been recommended to uncover residual hypoperfusion, and recent reviews [3,7] have highlighted the complementary role of CO_2_-derived indices in guiding resuscitation and risk assessment in critically ill patients. Importantly, in major abdominal surgery, the gap in P(v-a)CO_2_ has been shown to correlate with microcirculatory markers, and persistent elevation reflects inadequate perfusion despite apparently normal macrohemodynamic parameters. Several studies have demonstrated that P(v-a)CO_2_ is not only a physiological marker of CO_2_ clearance but also provides prognostic information regarding organ dysfunction and survival in surgical and critically ill populations. We have earlier reported that a P(v-a)CO_2_ value of 8.6 mmHg or higher, measured at 24 h post-surgery, significantly predicts 7-day mortality following major surgery in critically ill patients [8]. However, no studies have reported the relationship between changes in P(v-a)CO_2_ measured after surgery and postoperative patient prognosis. Thus, the present study was designed to evaluate the prognostic significance of changes in P(v-a)CO_2_ among critically ill patients after abdominal surgery, and to explore the relationship between ΔP(v-a)CO_2_ and other conventional microcirculatory markers, such as ScvO_2_, in order to better understand its potential role in postoperative risk stratification.

## 2. Materials and Methods

### 2.1. Study Design and Patient Enrollment

From September 2020 to October 2023, an observational study was conducted in the 22-bed ICU of a single tertiary hospital, and the data were retrospectively reviewed. All patients aged 18 years or older, admitted to the ICU after abdominal surgery under general anesthesia, were included, regardless of the surgical method (including laparotomy, laparoscopic surgery, or robotic surgery). We retrospectively included consecutive patients meeting these criteria, without additional exclusion related to the indication for catheterization, since both arterial and central venous catheters are inserted routinely as part of standard postoperative management in our ICU. Of the 86 patients included, 55 underwent elective or scheduled surgeries, and 31 underwent emergency surgeries. For the emergency surgery group, preoperative management was performed according to the Surviving Sepsis Campaign guidelines, including adequate hemodynamic resuscitation and timely administration of broad-spectrum antibiotics, in order to minimize the risk of insufficient preoperative preparation. In our ICU, invasive arterial and central venous catheter monitoring is routinely performed in all patients admitted after major abdominal surgery, regardless of intraoperative course or risk profile. Accordingly, all patients in this study had both a central venous catheter and an arterial line in place, followed by venous blood gas analysis (VBGA) and arterial blood gas analysis (ABGA) within 1 h (T0) and 24 h (T1) after ICU admission, respectively. Only patients with complete paired ABGA and VBGA results at both T0 and T1 were included in the final analysis to ensure internal consistency. As a result, patients without complete gas analysis data were excluded, which may introduce a risk of selection bias. However, since invasive arterial and central venous catheter monitoring is a routine practice in our ICU, we believe that this selection reflects standard clinical practice rather than selective catheter placement for research purposes. We retrospectively reviewed and analyzed data from patients who had complete ABGA and VBGA results at both T0 and T1. Patients were excluded if they met any of the following criteria: (1) under 18 years of age; (2) pregnancy; (3) death within 48 h after ICU admission, which made it impossible to collect data at both T0 and T1; (4) receipt of extracorporeal membrane oxygenation support therapy; (5) lack of an arterial or venous catheter owing to vessel stricture or thrombosis; (6) lack of venous or arterial gas analysis results at either T0 or T1; (7) preoperative ventilator care; (8) severe pulmonary dysfunction making general anesthesia unfeasible. Participants were divided into ΔP(v-a)CO_2_ ≤ 0 group and ΔP(v-a)CO_2_ > 0 group based on ΔP(v-a)CO_2_ variation, and demographic characteristics, fluid balance, hemodynamic values, and clinical outcomes were compared. A reference value of ≤6 mmHg for P(v-a)CO_2_ was adopted based on prior studies [3,4]. Although ROC analysis was conducted using P(v-a)CO_2_ values at T0 to assess the predictive value for postoperative ventilator care, the analysis did not yield a statistically significant cutoff for broader clinical outcomes such as mortality or complications, likely due to the limited sample size. However, the derived curve demonstrated a trend consistent with previous studies, including our prior work, thereby supporting the clinical relevance of the commonly referenced 6 mmHg threshold.

Accordingly, we performed subgroup analyses using this widely accepted cutoff value to ensure comparability with the existing literature and to explore potential associations between P(v-a)CO_2_ and clinical outcomes within our cohort. Consequently, participants were categorized into two groups based on their initial P(v-a)CO_2_ levels: low P(v-a)CO_2_ (≤6 mmHg) and high P(v-a)CO_2_ (>6 mmHg), and a subgroup analysis was conducted on these groups. In addition, for practical interpretation in line with our primary objective, patients were stratified into two groups (Δ ≤ 0 vs. Δ > 0), which allowed direct assessment of outcome differences according to the direction of change. This study received approval and monitoring from the Institutional Review Board of The Catholic University of Korea, Seoul St. Mary’s Hospital (No. IRB; KC24RISI0743) and adhered to the Declaration of Helsinki and its amendments.

### 2.2. Measurement of ΔP(v-a) CO_2_

The venous partial pressure of CO_2_ (PCO_2_) can be measured by obtaining mixed venous blood via a pulmonary artery catheter or central venous blood through a central venous catheter. However, pulmonary artery catheters are rarely utilized in our ICU, and previous studies [5,6] have demonstrated consistent results between the central venous-arterial PCO_2_ differences and the mixed venous-arterial PCO_2_ differences. Consequently, P(v-a)CO_2_ was calculated using PCO_2_ from the central venous blood in the current study. In our ICU, invasive arterial and central venous monitoring is performed as a routine practice for all patients requiring postoperative intensive care, according to institutional protocols. All participants were equipped with a central venous catheter and an arterial line either before or immediately upon admission to the ICU, and blood gas analysis was performed within 1 h (T0) and 24 h (T1) after admission. The central venous catheter was inserted via the Seldinger technique [7] after the internal jugular vein was visualized by ultrasound. Following the catheter insertion, a physician verified that the tip was positioned in the superior vena cava or upper right atrium by X-ray. The arterial lines were placed using a 22-gauge catheter (BD Angiocath Plus^®^, Becton Dickison Medical, Franklin Lakes, NJ, USA) in either the dorsalis pedis or radial arteries using aseptic techniques. Should securing an arterial line in these locations prove impossible, a 4-French catheter was placed in the femoral artery using the Seldinger technique. All ABGA and VBGA results were acquired using a point-of-care gas analyzer (ABL800 FLEX^®^, Radiometer, Copenhagen, Denmark) throughout the entire study period, thereby avoiding inter-device variability. To minimize bias affecting the results, the interval between the VBGA and ABGA was kept under 5 min, in accordance with our institutional protocol. All syringes used for blood sampling were pre-heparinized with the same balanced heparin solution, and blood gas analysis was performed immediately after collection to minimize temperature- and anticoagulant-related variability. P(v-a)CO_2_ was defined as follows and was calculated.P(v-a)CO_2_ = pCO_2_ of central venous blood − pCO_2_ of arterial blood

Also, ΔP(v-a)CO_2_ was defined and calculated as follows.ΔP(v-a)CO_2_ = P(v-a)CO_2_ measured at T1 − P(v-a)CO_2_ measured at T0

### 2.3. Data Collection and Study Outcomes

Clinical data were prospectively recorded in the electronic medical records and retrospectively reviewed for this study. The acquired information included demographics, fluid balances, underlying disease, and disease severity using APACHE II scores at ICU admission. Organ dysfunction was evaluated daily with SOFA scores starting from the time of ICU admission. Daily fluid balance, defined as the difference between total daily intake (including intravenous fluids, enteral fluids, medications, and blood products) and total output (encompassing losses through gastrointestinal, urinary, and other drains), ranged from 6 a.m. the previous day to 6 a.m. on the day of measurement. Nurses consistently recorded and calculated daily fluid balances using a specifically designed document. After conducting VBGA and ABGA, retrospective collection and recording of hemodynamic values such as lactate, central venous oxygen saturation (ScvO_2_), pH, bicarbonate, and P(v-a)CO_2_ were performed. The point in time to calculate the delta value is T1, with the delta value being defined as the difference between the values at T1 and T0.

Morbidity during postoperative hospitalization was monitored and recorded to determine clinical outcomes. Morbidity was classified according to the Clavien–Dindo classification, including only complications of grade III or higher in the analysis. Grade III complications were postoperative conditions that required endoscopic, radiological, or surgical intervention, whereas grade IV complications were life-threatening conditions that necessitated ICU management. Furthermore, grade V referred to a patient’s death following surgery. The primary outcome of the study was to determine prognosis differences based on changes in P(v-a)CO_2_ levels in critically ill patients experiencing an immediate post-abdominal surgery increase in P(v-a)CO_2_. The secondary outcome assessed the correlation between changes in P(v-a)CO_2_ and other conventional markers of microcirculation status in critically ill patients.

### 2.4. Acute Kidney Injury

Acute kidney injury (AKI) was characterized by any of the following criteria based on the Kidney Disease Improving Global Outcomes guidelines: (1) a urinary output of <0.5 mL/kg/hr for at least 6 h; (2) an increase in serum creatinine of ≥0.3 mg/dL(≥26.5 μmol/L) within 48 h; or (3) an increase in serum creatinine to ≥1.5 times the baseline within the previous 7 days [8].

### 2.5. Anastomosis Leakage

Defects in the integrity of the anastomosis wall, including suture and staple lines, can lead to communication with extra-luminal compartments. An abscess near the anastomosis site was classified as anastomotic leakage [9].

### 2.6. Bile Leakage

Bile leakage was defined as a bilirubin concentration in the drain fluid that is at least 3 times higher than the simultaneously measured serum bilirubin concentration on or after postoperative day 3, or by the need for radiologic or surgical intervention due to biliary collections or bile peritonitis [10].

### 2.7. Intra-Abdominal Fluid Collection

Intra-abdominal fluid collection encompasses both non-abscess abdominal fluid collection and abdominal abscess. A non-abscess intra-abdominal fluid collection is defined as a fluid accumulation measuring ≥3 cm in its largest dimension (length, width, or depth), as identified by computed tomography or ultrasonography, without signs of infection. An abdominal abscess is identified by a fluid collection detected through computed tomography or ultrasonography, with positive cultures from percutaneous drainage or reoperation [11].

### 2.8. Pleural Effusion

Pleural effusion was defined by the presence of interlobar fluid or blunting of the phrenico-costal angle on chest X-ray, based on a clear comparison between preoperative and postoperative images [12]. Chest X-rays were performed with the patient positioned in a semi-upright posture as much as possible. In addition, pleural effusion was routinely assessed using bedside ultrasound, which was performed as part of the institutional protocol for evaluating volume status.

### 2.9. Pneumonia

Postoperative pneumonia is characterized by the development of new consolidation, infiltration, or cavitation on postoperative chest X-ray, along with at least one of the following: (1) fever (>38 °C) without another identified cause, (2) leukocytosis (white blood cell count > 12 × 10^9^ L) or neutropenia (white blood cell count < 4 × 10^9^ L), and at least two of the following: (1) new or worsening cough, dyspnea, or tachypnea, (2) rales or bronchial breath sounds, (3) increased respiratory secretion, change in sputum character, new onset of purulent sputum, increased suction demand, or (4) gas exchange deterioration [13].

The treatment status and duration of treatment for patients undergoing mechanical ventilation or continuous renal replacement therapy were documented. The length of ICU stay, postoperative stay, and hospital stay were also recorded. Deaths during hospitalization were reviewed, and mortality rates at 7 days, 28 days, and during hospital stay post-surgery were analyzed.

### 2.10. Statistical Analysis

All statistical analyses were conducted using the SPSS statistical package software (version 24.0 for Windows; SPSS, Inc., Chicago, IL, USA). Based on the normal reference range of P(v-a)CO_2_, participants were divided into low PvaCO_2_ (≤6 mmHg) and high PvaCO_2_ (>6 mmHg) groups. We compared demographic characteristics, fluid balance, hemodynamic values, and clinical outcomes between the two groups. Continuous data are presented as mean ± standard deviation, and overall differences were tested using Student’s *t*-test or analysis of variance. The normal distribution of variables was assessed using the Kolmogorov–Smirnov test. Variables that were not normally distributed were analyzed using the Mann–Whitney test. Categorical variables were calculated using Fisher’s exact test or the Chi-squared (χ^2^) test. The relationship between the ΔP(v-a)CO_2_ and various parameters measured at T1 and T0 was assessed using partial correlation, linear regression analysis, and Bland–Altman plots. Differences were considered statistically significant for *p*-values of <0.05. Multivariate logistic regression analysis was performed to identify predisposing factors for 28-day mortality and postoperative mechanical ventilator support using significant variables from the univariate analysis.

## 3. Results

During the study period, a total of 1275 patients were admitted to the surgical ICU post-surgery and enrolled in the study. Due to the retrospective design of the study, VBGA was not routinely performed until a certain time point. After its implementation as part of routine clinical practice, most patients from the earlier period were excluded because of missing ABGA/VBGA data. So, 1190 were excluded due to lack of ABGA or VBGA records at T0 or T1. Consequently, 86 patients were analyzed, as shown in Figure 1. The primary outcome of this study was the need for postoperative mechanical ventilation. Secondary outcomes included CRRT use, acute kidney injury (AKI), other postoperative morbidities, ICU/hospital stay, and mortality.

Participants were divided into ΔP(v-a)CO_2_ ≤ 0 group (*n* = 49, 57%) and ΔP(v-a)CO_2_ > 0 group (*n* = 37, 43%), and their demographic characteristics, fluid balance, hemodynamic values, and clinical outcomes were compared (Table 1).

No significant differences were found in demographics such as age, sex, body mass index (BMI), and underlying diseases between the ΔP(v-a)CO_2_ ≤ 0 group and the ΔP(v-a)CO_2_ > 0 group. Disease severity, assessed by the total SOFA score at T0 (3.7 ± 2.9 vs. 2.9 ± 2.8, *p = 0.243*), and septic shock status at T0 (*n* = 8, 16.3% vs. *n* = 2, 5.4%, *p = 0.117*), were also comparable between the two groups. Similarly, the delta value of the total SOFA score did not differ. Regarding fluid balance, no significant differences were noted in total intake and output amounts on both the day of surgery and the following day. In terms of hemodynamic parameters, there were no significant differences between the two groups in lactate levels at T0, Δlactate at T1, ScvO_2_ at T0, and ScvO_2_ at T1. The value of P(v-a)CO_2_ at T0 was significantly higher in the ΔP(v-a)CO_2_ ≤ 0 group (10.6 ± 3.8 vs. 4.5 ± 3.3, *p < 0.001*). However, P(v-a)CO_2_ at T1 (5.9 ± 2.8 vs. 9.5 ± 4.3, *p < 0.001*) and ΔP(v-a)CO_2_ at T1 (−4.6 ± 3.8 vs. 5 ± 4.3, *p < 0.001*) were significantly lower in the ΔP(v-a)CO_2_ ≤ 0 group than in the ΔP(v-a)CO_2_ > 0 group. Regarding primary outcome, in the overall study population of 86 patients who were eligible for analysis, the need for postoperative mechanical ventilation occurred in 19/86 patients (22%), with no significant difference between the two groups (*n* = 12, 24.5% vs. *n* = 7/37, 18.9%, *p = 0.607*). For secondary outcomes, no significant differences were found in CRRT use, AKI, ICU stay, or mortality between the two groups. During the study period, mortality occurred in 5 patients (5.8%): 7-day (*n* = 1, 1.2%), 28-day (*n* = 4, 4.7%), and in-hospital (*n* = 5, 5.8%), without significant group differences.

### Patients with P(v-a)CO_2_ at T0 Greater than 6 mmHg

For subgroup analysis, the analysis was selectively re-performed only in participants with high P(v-a)CO_2_(>6 mmHg) at T0 (Table 2).

The subjects of analysis were the ΔP(v-a)CO_2_ ≤ 0 group (*n* = 45, 80.4%) and ΔP(v-a)CO_2_ > 0 group (*n* = 11, 19.6%). There were no significant differences in demographics or disease severity, as represented by the total SOFA score, between the two groups. In terms of hemodynamic parameters, no significant differences were observed in lactate and ScvO_2_ levels at T0 between the two groups. Furthermore, the proportion of patients who underwent laparoscopic or robotic surgery did not differ significantly between the two groups (12 of 45 patients [26.7%] in the ΔP(v-a)CO_2_ ≤ 0 group vs. 3 of 11 patients [27.3%] in the ΔP(v-a)CO_2_ > 0 group). Among these patients, P(v-a)CO_2_ values at T0 (10.6 ± 3.6 vs. 10.4 ± 2.7 mmHg, *p* = 0.845) and T1 (7.6 ± 3.9 vs. 7.0 ± 5.0 mmHg, *p* = 0.652), as well as ΔP(v-a)CO_2_ (−3.0 ± 5.3 vs. −3.4 ± 5.8 mmHg, *p* = 0.818), showed no significant differences between groups. However, lactate levels at T1 (2.5 ± 1.6 vs. 4.8 ± 5.2, *p = 0.011*) and Δlactate at T1(−0.9 ± 2.4 vs. 1.5 ± 2.8, *p = 0.006*) were significantly lower in the ΔP(v-a)CO_2_ ≤ 0 group than in the ΔP(v-a)CO_2_ > 0 group. The initial P(v-a)CO_2_ value at T0 was significantly higher in the ΔP(v-a)CO_2_ ≤ 0 group (11 ± 3.5 vs. 8.6 ± 1.8, *p < 0.036*). However, P(v-a)CO_2_ at T1(6.2 ± 2.8 vs. 12.6 ± 5.3, *p < 0.003*) and ΔP(v-a)CO_2_ at T1(−4.8 ± 3.8 vs. 3.9 ± 5.2, *p < 0.001*) were significantly lower in the ΔP(v-a)CO_2_ ≤ 0 group than in the ΔP(v-a)CO_2_ > 0 group. In contrast to the overall study population, patients with P(v-a)CO_2_ > 6 mmHg at T0 showed that the incidence of mechanical ventilation was significantly lower in the ΔP(v-a)CO_2_ ≤ 0 group (*n* = 10, 22.2%) than in the ΔP(v-a)CO_2_ > 0 group (*n* = 7, 54.5%, *p = 0.033*). The duration of mechanical ventilation was also shorter (0.9 ± 2.2 vs. 3.6 ± 5.6 days, *p = 0.011*).

Regarding secondary outcomes, CRRT was required less frequently in the ΔP(v-a)CO_2_ ≤ 0 group (*n* = 4, 8.9%) compared to the ΔP(v-a)CO_2_ > 0 group (*n* = 4, 36.4%, *p = 0.020*). The duration of CRRT was also shorter (0.5 ± 1.6 vs. 2.8 ± 5.6 days, *p = 0.016*). The incidence of AKI was significantly lower (*n* = 5, 11.1% vs. *n* = 4, 36.4%, *p = 0.041*). ICU stay was shorter in the ΔP(v-a)CO_2_ ≤ 0 group, though hospital stay did not differ (Table 3).

During the study period, mortality occurred in 4 cases (7.1%) among all participants, with 7-day, 28-day, and in-hospital mortality rates of 1 (1.8%), 4 (7.1%), and 4 patients (7.1%), respectively. Among these, only the 7-day mortality rate was significantly lower in the ΔP(v-a)CO_2_ ≤ 0 group (*n* = 0, 0% vs. *n* = 1, 9.1%, *p = 0.041*).

Partial correlation analysis was conducted to assess the association between ΔP(v-a)CO_2_ and various hemodynamic parameters measured at T1 and T0, with adjustments for age and sex (Table 4).

Among the parameters, ScvO_2_ measured at T1 exhibited a negative correlation with ΔP(v-a)CO_2_ (partial correlation coefficients: −0.273, *p = 0.045*) and was also negatively correlated with ΔP(v-a)CO_2_ (β ± SE: −0.522 ± −0.221, correlation coefficients: −0.305, *p = 0.022*) in the linear regression analysis (Table 5).

Figure 2 further demonstrates the negative correlation between ScvO_2_ measured at T1 and ΔP(v-a)CO_2._

Table 6 displays the outcomes of the logistic regression analysis concerning postoperative 28-day mortality.

Following univariate analysis, significant risk factors identified included total SOFA scores at T0 and T1, Δbicarbonate at T1, lactate levels at T1, Δlactate at T1, and ScvO_2_ levels at T1. However, no significant risk factors emerged in predicting postoperative 28-day mortality following multivariate analysis. Regarding postoperative mechanical ventilator treatment, significant variables identified after univariate analysis included changes in ΔP(v-a)CO_2_ status, increases in ΔSOFA scores at T1, and decreases in ScvO_2_ levels at T1, as shown in Table 7.

Among these, increased ΔSOFA scores at T1 and decreased ScvO_2_ levels at T1 were confirmed as significant risk factors for postoperative mechanical ventilator treatment [[2], and (OR = 0.898, 95% CI: 0.819–0.985, *p = 0.022*), respectively].

## 4. Discussion

Our results indicated that ΔP(v-a)CO_2_ values measured at T1, 24 h after ICU admission, were significantly correlated with ScvO_2_ levels at T1, suggesting an association with tissue perfusion status. Among patients in the high-risk group with P(v-a)CO_2_ values exceeding 6 mmHg at T0, those with ΔP(v-a)CO_2_ values of 0 mmHg or lower at T1 had significantly better clinical outcomes, including reduced duration of mechanical ventilator and CRRT use, shorter ICU stays, lower incidence of postoperative AKI, and reduced 7-day mortality rate.

Maintaining adequate tissue perfusion and oxygenation in patients who have undergone major surgery is essential to preserve organ function and surgical site integrity. ScvO_2_ has been used as one of the various tissue perfusion parameters in clinical settings. It is known to reliably reflect changes in oxygen delivery and tissue oxygen consumption [14], and a previous study reported that close monitoring of tissue O_2_ extraction using ScvO_2_ measurements can reduce postoperative organ dysfunction through the early detection and correction of abnormal tissue oxygenation [15]. Unfortunately, in this study, the ScvO_2_ level measured at T0 did not show a clear correlation with clinical outcomes. However, the reduced ScvO_2_ level at T1 was a significant risk factor for postoperative mechanical ventilation after multiple logistic regression analysis. A decrease in ScvO_2_ generally indicates increased metabolic demands, especially when oxygen extraction may be abnormally increased for oxygen uptake in cases of extensive tissue injury such as major surgery or multiple trauma. Additionally, an abnormally decreased ScvO_2_ can indicate hypovolemia or anemia due to intraoperative hemorrhage or insufficient oxygen supply from decreased cardiac output. Therefore, patients with reduced ScvO_2_ levels would require early correction of the imbalance between oxygen delivery and consumption, and they may require more aggressive ventilator therapy to improve oxygen supply, in addition to general conservative oxygen support [16]. In particular, patients who underwent major abdominal surgery, the focus of our study, often experienced intraoperative bleeding or early hemodynamic instability when admitted to the ICU after surgery. Additionally, decreased hemoglobin or cardiac output can result in inadequate oxygen supply to tissues and vital organs, leading to vital organ impairment. Moreover, respiratory muscle movement is often restricted by postoperative pain and residual anesthetic effects, leading to frequent asynchrony immediately after surgery. This asynchrony promotes atelectasis and increased dead space in the immediate postoperative period, ultimately leading to increased ventilation/perfusion mismatch. Furthermore, O_2_ extraction often increases in response to the elevated oxygen demand for recovery from damage caused by extensive tissue manipulation during surgery or for healing at the anastomosis site. Since the subjects of this study were limited to postoperative patients with high risks such as dynamic imbalance of respiratory muscles or increased O_2_ extraction from tissues, differences in clinical outcomes might be observed based on ScvO_2_ values. We anticipate that the clinical application of ScvO_2_ could be more beneficial in monitoring the condition of postoperative patients and improving clinical outcomes compared to other patient groups. Furthermore, physicians should be aware that the ScvO_2_ value measured 24 h post-surgery in patients who have undergone major abdominal operations could indicate a risk factor for requiring postoperative mechanical ventilation. If the ScvO_2_ value decreases, the patient’s condition should be closely monitored, including cardiac output, accurate metabolic demands, and volume status; physicians should also promptly provide adequate fluid therapy and administer vasopressors.

Although ScvO_2_ is often used as an indicator of global oxygen balance, its role as a direct marker of microcirculatory function remains limited. In certain clinical conditions, such as sepsis or septic shock, elevated ScvO_2_ levels may reflect impaired oxygen extraction due to microcirculatory shunting or mitochondrial dysfunction, which can limit its reliability in accurately reflecting tissue perfusion. As noted in a multicenter study by Pope et al., the interpretation of ScvO_2_ should consider the clinical context, as overly high ScvO_2_ values may reflect impaired oxygen utilization rather than adequate perfusion [17]. Nonetheless, compared to lactate, which typically reflects delayed metabolic consequences of hypoperfusion, ScvO_2_ and ΔP(v-a)CO_2_ may offer more immediate insights into circulatory adequacy and oxygen dynamics. While each parameter has its own limitations, the combined interpretation of ScvO_2_ and ΔP(v-a)CO_2_ may enhance the early detection of perfusion mismatch and allow more timely intervention. In postoperative patients with high metabolic demands and fluctuating volume status, this can be particularly beneficial. Thus, our findings suggest that complementary use of these markers may provide additional insights alongside conventional parameters such as lactate. Impaired microcirculation disorders can result from increased blood viscosity, endothelial dysfunction, or glycocalyx degradation associated with severe tissue injury or sepsis [18], or from microthrombi formation associated with disseminated intravascular coagulation. In our study population—comprising patients who underwent major abdominal surgery—large-volume fluid administration is commonly required, which may lead to hemodilution. This hemodilution can, in some cases, attenuate the viscosity-related component of microcirculatory impairment. However, despite potentially preserved macro-hemodynamic parameters such as blood pressure, perfusion to peripheral organs may still be compromised. Therefore, macro-circulation variables may not reliably reflect the underlying status of the microcirculation in such clinical settings. In such instances, although blood pressure might be maintained, perfusion to major organs in the periphery could be compromised, hence macro-circulation variables may not accurately reflect the real microcirculation status. Moreover, the glycocalyx, which possesses a net negative charge, functions as an endothelial barrier by forming a wall for negatively charged proteins in the plasma, thereby helping to maintain osmotic pressure toward the blood vessel’s lumen. When plasma protein leaks into the interstitium due to glycocalyx degradation, tissue perfusion does not improve as expected after sufficient fluid resuscitation because oncotic pressure cannot be maintained [18,19]. Therefore, additional biological markers that can more accurately represent tissue perfusion are needed in these cases. CO_2_ serves as an indicator of tissue perfusion because it is about 20 times more soluble in water than oxygen. In ischemic tissues with poor perfusion, oxygen deficiency triggers anaerobic respiration for energy production, which also increases CO_2_ production as a by-product. Consequently, the increased CO_2_ is likely to diffuse into the venous effluent due to its higher solubility in water, making CO_2_ in venous blood an effective indicator of tissue hypoperfusion. The results of this study reveal that in the high-risk group, a P(v-a)CO_2_ measured at T1 greater than zero with an increase over 6 mmHg post-admission correlates with increased durations of mechanical ventilation and CRRT usage, extended ICU stays, and elevated incidences of postoperative AKI and 7-day mortality rate. Gustavo et al. [20] reported that persistently elevated P(v-a)CO_2_ values in patients newly diagnosed with septic shock correlate with severe multi-organ dysfunction and 28-day mortality. In our previous study published in 2024 [21], we established that a P(v-a)CO_2_ value of 8.6 mmHg or higher, measured 24 h post-admission, served as a significant prognostic indicator for 7-day mortality among ICU patients after abdominal surgery. Consequently, based on prior research and our findings, in patients entering the ICU post-major surgery, a P(v-a)CO_2_ level exceeding 6 mmHg immediately post-admission should alert physicians to possible decreases in tissue perfusion, serial measurements of P(v-a)CO_2_ may be helpful in monitoring perfusion changes, though prospective studies are needed. Furthermore, the authors anticipate that whether timely and aggressive intervention based on these findings improves outcomes requires confirmation in future prospective trials.

Interestingly, our findings showed that the ScvO2 value measured at T1, indicating tissue perfusion, demonstrated a negative correlation with ΔP(v-a)CO_2_. Typically, an increase in ScvO_2_ suggests enhanced oxygen delivery, whereas a decrease in P(v-a)CO_2_ signals improved tissue perfusion and CO_2_ clearance. Consequently, ScvO_2_ and P(v-a)CO_2_ are interrelated and often assessed together to evaluate tissue perfusion in critically ill patients. Ospina et al. [1] observed that low ScvO_2_ (<70%) and high P(v-a)CO_2_ (>6 mmHg) correlate with adverse outcomes in septic shock patients. Moreover, several studies show that ScvO_2_ and P(v-a)CO_2_ are inversely proportional. Another study [5] also found that increased P(v-a)CO_2_ in septic shock patients corresponded with reduced ScvO_2_ levels and poor tissue perfusion. Additionally, successful resuscitation in septic shock patients led to a significant rise in ScvO_2_ and a decline in P(v-a)CO_2_ values. These findings are consistent with our study results, which demonstrated a negative correlation between ScvO_2_ levels and the change in P(v-a)CO_2_ values at T1. These findings suggest that a combined interpretation of ScvO_2_ and P(v-a)CO_2_ may help physicians gain a more nuanced understanding of hemodynamic status in patients with compromised tissue perfusion, although further prospective studies are warranted to determine its role in guiding interventions.

Despite these interesting results, the findings of the current study should be interpreted with caution due to various limitations. First, since it uses a retrospective design, selection bias cannot be eliminated, and decisions regarding interventions and treatments were not randomized. However, all patients admitted to the ICU were managed by a single intensivist, and no significant changes in critical management principles were observed. Thus, we anticipate that this may reduce potential biases in treatment strategies. In addition, clinically important perioperative and intensive care variables such as transfusion, use of vasopressors/inotropes, and anesthesia duration were not consistently available in this retrospective dataset, which may have influenced both ScvO_2_ and CO_2_ clearance. Nevertheless, in our cohort, fluid balance, total intake, IV intake, and output showed no significant differences between the groups, supporting the validity of our analysis of ΔP(v-a)CO_2_ and ΔScvO_2_ changes between T0 and T1. Second, we analyzed only ScvO_2_ and P(v-a)CO_2_ values measured immediately and within 24 h after ICU admission. Although serial ScvO_2_ and P(v-a)CO_2_ measurements were performed several times, there were insufficient participant numbers for analysis at other time points. These values are vital not only at specific times post-major surgery but also as tools to evaluate responses. In the future, the clinical significance of ScvO_2_ and P(v-a)CO_2_ values at other times should be confirmed. Thirdly, a 24 h interval may be too long to detect trends and critical time points in P(v-a)CO_2_ changes. In our study, we performed timely checks of P(v-a)CO_2_ when the patient’s condition was unstable post-surgery, but the analysis was unsuccessful due to insufficient participant numbers. In the future, the clinical significance of short-term changes in P(v-a)CO_2_ should be verified. Fourthly, among the patients whose gas analysis was performed post-surgery, those who underwent both laparoscopic and robotic surgeries were included. Therefore, it is important to consider the transient rise in PaCO_2_ caused by CO_2_ pneumoperitoneum during these procedures. However, several studies have reported that in patients without preexisting pulmonary disease, acid–base imbalances and elevated PaCO_2_ levels caused by CO_2_ pneumoperitoneum generally normalize within a few hours after extubation and surgery. In our study, since the majority of patients underwent major abdominal surgery under general anesthesia, individuals with severe preoperative pulmonary dysfunction were excluded, and thus the influence of pneumoperitoneum on postoperative PaCO_2_ levels is presumed to be minimal [22]. Since our study focused on the P(v-a)CO_2_ value—an indirect indicator of tissue perfusion—and this was measured 24 h after ICU admission, the potential influence of intraoperative PaCO_2_ elevation is expected to be minimal. Fifthly, due to the retrospective design of the study, the exact temporal relationship between ΔP(v-a)CO_2_ measurements and clinical interventions such as mechanical ventilation or CRRT initiation could not be consistently confirmed. As a result, it is unclear whether changes in P(v-a)CO_2_ preceded or followed clinical deterioration or therapeutic actions. Nevertheless, since the majority of patients were already on mechanical ventilation at T0 (*n*: 18, 94.7%), we believe that the observed ΔP(v-a)CO_2_ values still reflect meaningful physiological changes rather than being solely post-intervention effects. Sixthly, this study has several limitations. Being a retrospective, single-center study with a small sample size—especially in the ΔP(v-a)CO_2_ > 0 group (*n* = 11)—it may have limited statistical power to detect significant differences in key clinical outcomes such as mechanical ventilation, AKI, CRRT use, and 7-day mortality. Post hoc power analysis using G*Power (version 3.1.9.7) revealed that the achieved power for most outcomes was below 0.8, suggesting insufficient power to draw definitive conclusions. Additionally, the retrospective nature of the study and the relatively small sample size—composed mainly of patients undergoing elective surgeries—likely contributed to the low incidence of AKI and CRRT use, further limiting the statistical power and generalizability of our findings. Therefore, larger prospective studies are warranted to validate our results and further clarify the prognostic value of ΔP(v-a)CO_2_ in this population. Finally, interpreting P(v-a)CO_2_ values requires a deeper analysis of various factors that affect the relationship between pCO_2_ and CO_2_ content in the blood. However, environmental conditions in our ICU, such as humidity between 35 and 40% and a temperature of 24 °C, were kept consistent to reduce these errors. Gas analysis was conducted using a precision-managed, point-of-care gas analyzer in the ICU. Nevertheless, this study offers distinct implications from previous studies. Our study demonstrated a significant correlation between ScvO_2_ levels and changes in P(v-a)CO_2_ values. These findings offer deeper insights into tissue perfusion. The relevance of this study is underscored by its focus on patients post-major surgery who are at a high risk of fatal complications from inadequate tissue perfusion. However, we were unable to demonstrate a statistical difference in postoperative morbidity, as the sample size was insufficient. In the near future, we aim to conduct a well-designed prospective randomized controlled study with a larger cohort, particularly focusing on patients with more severe conditions.

In conclusion, fluctuations in P(v-a)CO_2_ levels demonstrated a modest inverse association with ScvO_2_ and may serve as supportive markers for postoperative risk stratification in critically ill patients following major abdominal surgery. However, given the retrospective design, relatively small sample size, and the inclusion of only patients with invasive monitoring, these findings should be considered hypothesis-generating rather than definitive. Therefore, further validation in well-designed, prospective multicenter studies with larger cohorts is required to establish the prognostic utility of ΔP(v-a)CO_2_ in postoperative critical care.

## Figures and Tables

**Figure 1 biomedicines-13-02295-f001:**
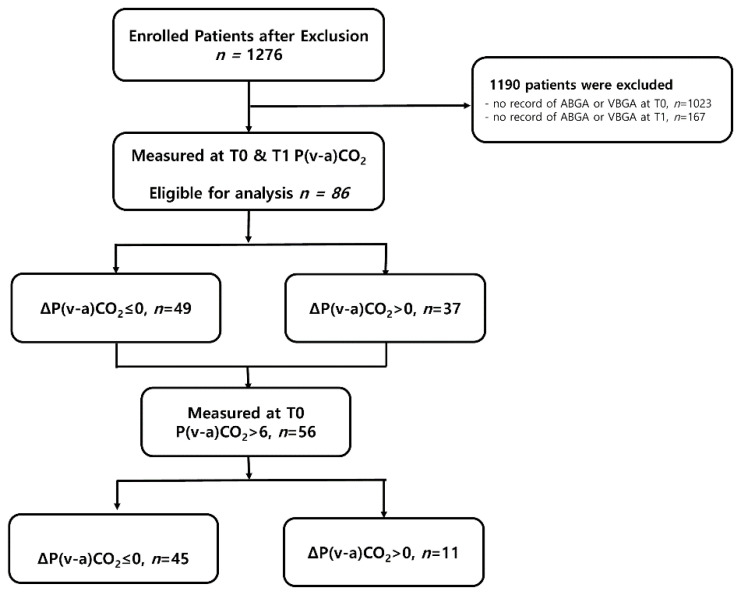
Schematic diagram of study enrollment. ABGA: arterial blood gas analysis, P(v-a)CO_2_: venous-to-arterial carbon dioxide partial pressure difference, T0: 1 h after ICU admission, T1: 24 h after ICU admission, VBGA: venous blood gas analysis.

**Figure 2 biomedicines-13-02295-f002:**
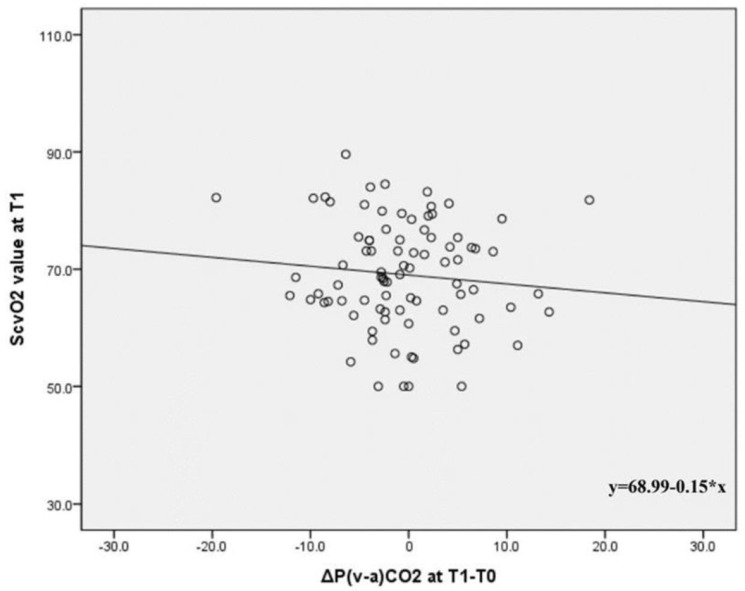
Correlation between ScvO2 at T1 and ΔP(v-a)CO_2_ at T1-T0. P(v-a)CO_2_: venous-to-arterial carbon dioxide partial pressure difference, ScvO_2_: central venous oxygen saturation, T0: 1 h after ICU admission, T1: 24 h after ICU admission.

**Table 1 biomedicines-13-02295-t001:** Comparative analysis of demographic characteristics, fluid balance, hemodynamic value and clinical outcomes according to ΔP(v-a)CO_2_.

Variables	ΔP(v-a)CO_2_ ≤ 0 (*n* = 49)	ΔP(v-a)CO_2_ > 0 (*n* = 37)	*p*-Value
Age, years	67.5 (31–96)	70.2 (30–85)	*0.339*
Sex, male, *n* (%)	18 (36.7%)	19 (51.4%)	*0.194*
BMI (Kg/m^2^)	23.7 ± 3.9	23.3 ± 3.3	*0.613*
APACHE II (mean, ± SD)	13.8±6.5	14.6±5.8	*0.56*
Total SOFA score at T0 (mean, ±SD)	3.7±2.9	2.9±2.8	*0.243*
Total SOFA score at T1 (mean, ±SD)	3.6±3.3	3.3±3.2	*0.699*
ΔSOFA score at T1-T0 (mean, ±SD)	−0.1 ± 1.8	0.3±2.3	*0.3*
Total SOFA score at T2 (mean, ±SD)	2.6±2.9	3.4±4.2	*0.317*
Total SOFA score at T3 (mean, ±SD)	2.2±2.5	2.8±3.5	*0.389*
Sepsis status at T0	29 (59.2%)	18 (48.6%)	*0.385*
Septic shock status at T0	8 (16.3%)	2 (5.4%)	*0.117*
Underlying disease, *n* (%)			
Diabetes mellitus	9 (18.4%)	11 (29.7%)	*0.303*
HBP	22 (44.9%)	19 (51.4%)	*0.664*
CVA	9 (18.4%)	0	*0.009*
Liver cirrhosis	2 (4.1%)	1 (2.7%)	*1.000*
Chronic renal failure	5 (10.2%)	4 (10.8%)	*1.000*
**Intake and Output (mean, ±SD)**			
Total intake at POD 0, mL	3046.1±1626.5	2887.6±1266.5	*0.625*
Total output at POD 0, mL	2108.2±1091.5	1777.9±966.7	*0.148*
Total intake at POD 1, mL	3316.1±944.1	3242.5±976.5	*0.725*
Total output at POD 1, mL	2326.2±906.2	2235.2±905.1	*0.646*
**Hemodynamic value (mean, ±SD)**			
Lactate at T0, mmol/L	3.3±2.7	3.1±2.2	*0.617*
Lactate at T1, mmol/L	2.4±1.6	2.9±3.1	*0.327*
ΔLactate at T1-T0, mmol/L	−0.9 ± 2.4	−0.2 ± 2.3	*0.13*
ScvO_2_ at T0, %	66.1±9.7	68.4±8.8	*0.249*
ScvO_2_ at T1, %	69±9.5	69.1±8.9	*0.949*
pH of ABGA at T0	7.38±0.06	7.37±0.05	*0.135*
pH of ABGA at T1	7.42±0.05	7.4±0.05	*0.116*
ΔpH at T1-T0	0.03 ± 0.06	0.03 ± 0.08	*0.931*
Bicarbonate at T0, mmol/L	21.4±4	21.3±3.1	*0.947*
Bicarbonate at T1, mmol/L	24.9±3.8	22.3±4	*0.002*
ΔBicarbonate at T1-T0, mmol/L	3.6 ± 3.9	1±3.7	*0.002*
P(v-a)CO_2_ at T0, mmHg	10.6±3.8	4.5±3.3	*<0.001*
P(v-a)CO_2_ at T1, mmHg	5.9±2.8	9.5±4.3	*<0.001*
ΔP(v-a)CO_2_ at T1-T0, mmHg	−4.7 ± 3.8	5±4.3	*<0.001*
**Clinical outcomes**			
Use of mechanical ventilation, *n* (%)	12 (24.5%)	7 (18.9%)	*0.607*
Use of CRRT, *n* (%)	5 (10.2%)	7 (18.9%)	*0.348*
Length of mechanical ventilation, day	0.9±2.1	1.2±3.4	*0.624*
Length of CRRT, day	0.5±1.7	1±3.3	*0.343*
Length of ICU stay, day	2.8±2.3	3.2±3.5	*0.534*
Length of postoperative stay, day	15±10.9	13.2±9.6	*0.431*
Length of hospital stay, day	20.4±13.3	16.3±10.5	*0.124*
Postoperative morbidities			
Acute kidney injury	6 (12.2%)	7 (18.9%)	*0.545*
Anastomosis leakage	1 (2%)	1 (2.7%)	*1.000*
Bile leakage	1 (2%)	0	*1.000*
Intra-abdominal fluid collection	2 (4.1%)	2 (5.4%)	*1.000*
Pleural effusion/Pneumonia	3 (6.1%)	4 (10.8%)	*0.457*
7 days mortality, *n* (%)	0	1 (2.7%)	*0.43*
28 days mortality, *n* (%)	2 (4.1%)	2 (5.4%)	*1.000*
In-hospital mortality, *n* (%)	2 (4.1%)	3 (8.1%)	*0.648*

ABGA: arterial blood gas analysis, APACHE II: acute physiology and chronic health evaluation II, BMI: body mass index, CRRT: continuous renal replacement therapy, CVA: cerebrovascular accident, HBP: high blood pressure, ICU: intensive care unit, POD: postoperative day, P(v-a)CO_2_: venous-to-arterial carbon dioxide partial pressure difference, ScvO_2_: central venous oxygen saturation, SD: standard deviation, SOFA: sequential organ failure assessment, T0: within 1 h after admission to intensive care unit, T1: within 24 h after admission to intensive care unit.

**Table 2 biomedicines-13-02295-t002:** Comparative analysis of demographic characteristics, fluid balance and hemodynamic value according to ΔP(v-a)CO_2_ in patients with P(v-a)CO_2_ at T0 > 6 mmHg.

Variables	ΔP(v-a)CO_2_ ≤ 0 (*n* = 45)	ΔP(v-a)CO_2_ > 0 (*n* = 11)	*p*-Value
Age, years	67.8 (31–96)	74 (56–85)	*0.139*
Sex, male, *n* (%)	17 (37.8%)	7 (63.6%)	*0.176*
BMI (Kg/m^2^)	23.9 ± 3.8	24.2 ± 2.4	*0.854*
APACHE II (mean, ± SD)	13.5±6.5	17.9±6.7	*0.051*
Total SOFA score at T0 (mean, ±SD)	3.5±2.7	4.9±3.8	*0.152*
Total SOFA score at T1 (mean, ±SD)	3.4±3.3	5.7±4.1	*0.054*
ΔSOFA score at T1-T0 (mean, ±SD)	−0.1 ± 1.8	0.8±3.4	*0.447*
Total SOFA score at T2 (mean, ±SD)	2.4±2.8	6.8±5.7	*0.032*
Total SOFA score at T3 (mean, ±SD)	2±2.3	4.9±4.9	*0.085*
Sepsis status at T0	25 (55.6%)	8 (72.7%)	*0.496*
Septic shock status at T0	8 (17.8%)	1 (9.1%)	*0.671*
**Underlying disease, *n* (%)**			
Diabetes mellitus	6 (13.3%)	2 (18.2%)	*0.649*
HBP	20 (44.4%)	6 (54.5%)	*0.738*
CVA	8 (17.8%)	0	*0.333*
Liver cirrhosis	1 (2.2%)	0	*1.000*
Chronic renal failure	3 (6.7%)	2 (18.2%)	*0.251*
**Intake and Output (mean, ±SD)**			
Total intake at POD 0, mL	3025.1±1636.9	3864.9±1502.2	*0.127*
Total output at POD 0, mL	2085.5±1086.3	1806.9±385.5	*0.213*
Total intake at POD 1, mL	3335.9±979.9	3540.5±1220.4	*0.557*
Total output at POD 1, mL	2317±935.8	1979.4±873.3	*0.282*
**Hemodynamic value (mean, ±** **SD)**			
Lactate at T0, mmol/L	3.3±2.7	3.3±2.9	*0.985*
Lactate at T1, mmol/L	2.5±1.6	4.8±5.2	*0.011*
ΔLactate at T1-T0, mmol/L	−0.9 ± 2.4	1.5±2.8	*0.006*
ScvO_2_ at T0,%	66.5±9.6	62.4±7.8	*0.195*
ScvO_2_ at T1,%	69.2±9.2	64.1±8.4	*0.101*
pH of ABGA at T0	7.38±0.06	7.37±0.05	*0.583*
pH of ABGA at T1	7.41±0.05	7.41±0.07	*0.894*
ΔpH at T1-T0	0.03 ± 0.06	0.04 ± 0.08	*0.693*
Bicarbonate at T0, mmol/L	21.3±3.9	20±3.2	*0.324*
Bicarbonate at T1, mmol/L	24.8±3.5	20.6±3.4	*0.001*
ΔBicarbonate at T1-T0, mmol/L	3.5 ± 3.4	0.6±3.9	*0.018*
P(v-a)CO_2_ at T0, mmHg	11±3.5	8.6±1.8	*0.036*
P(v-a)CO_2_ at T1, mmHg	6.2±2.8	12.6±5.3	*0.003*
ΔP(v-a)CO_2_ at T1-T0, mmHg	−4.8 ± 3.8	4±5.2	*<0.001*

ABGA: arterial blood gas analysis, APACHE II: acute physiology and chronic health evaluation II, BMI: body mass index, CVA: cerebrovascular accident, HBP: high blood pressure, POD: postoperative day, P(v-a)CO_2_: venous-to-arterial carbon dioxide partial pressure difference, ScvO_2_: central venous oxygen saturation, SD: standard deviation, SOFA: sequential organ failure assessment, T0: within 1 h after admission to intensive care unit, T1: within 24 h after admission to intensive care unit.

**Table 3 biomedicines-13-02295-t003:** Comparative analysis of clinical outcomes according to ΔP(v-a)CO_2_ in patients with P(v-a)CO_2_ at T0 > 6 mmHg.

Variables	ΔP(v-a)CO_2_ ≤ 0 (*n* = 45)	ΔP(v-a)CO_2_ > 0 (*n* = 11)	*p*-Value
**Clinical outcomes**			
Use of mechanical ventilation, *n* (%)	10 (22.2%)	7 (54.5%)	*0.033*
Use of CRRT, *n* (%)	4 (8.9%)	4 (36.4%)	*0.020*
Length of mechanical ventilation, day	0.9±2.2	3.6±5.6	*0.011*
Length of CRRT, day	0.5±1.6	2.8±5.6	*0.016*
Length of ICU stay, day	2.8±2.4	5.5±5.4	*0.014*
Length of postoperative stay, day	15.1±11.3	13.9±8.9	*0.744*
Length of hospital stay, day	20.6±13.6	16.2±8.8	*0.307*
Postoperative morbidities			
Acute kidney injury	5 (11.1%)	4 (36.4%)	*0.041*
Anastomosis leakage	1 (2.2%)	1 (9.1%)	*0.357*
Bile leakage	1 (2.2%)	0	*1.000*
Intra-abdominal fluid collection	2 (4.4%)	1 (9.1%)	*0.488*
Pleural effusion/Pneumonia	2 (4.4%)	1 (9.1%)	*0.488*
7 days mortality, *n* (%)	0	1 (9.1%)	*0.041*
28 days mortality, *n* (%)	2 (4.4%)	2 (18.2%)	*0.113*
In-hospital mortality, *n* (%)	2 (4.4%)	2 (18.2%)	*0.113*
Composite mortality, *n* (%)	2(4.4%)	2(18.2%)	*0.113*

CRRT: continuous renal replacement therapy, ICU: intensive care unit.

**Table 4 biomedicines-13-02295-t004:** Partial correlation between ΔP(v-a)CO_2_ and various parameters measured at T1 and T0 after adjustment for age and sex.

Parameters	Partial Correlation Coefficients	*p*-Value
ΔSOFA score at T1-T0	−0.136	*0.325*
ΔLactate at T1-T0	0.133	*0.339*
Total SOFA at T0	0.206	*0.135*
Total SOFA at T1	0.093	*0.503*
Lactate level at T0	−0.015	*0.912*
Lactate level at T1	0.106	*0.444*
ScvO2 at T0	−0.13	*0.349*
ScvO2 at T1	−0.273	*0.045*

P(v-a)CO_2_: venous-to-arterial carbon dioxide partial pressure difference, SOFA: sequential organ failure assessment, T0: within 1 h after admission to intensive care unit, T1: within 24 h after admission to intensive care unit.

**Table 5 biomedicines-13-02295-t005:** Linear regression analysis between ΔP(v-a)CO_2_ and various parameters measured at T1 and T0.

Parameters	β ± SE	Correlation Coefficients	*p*-Value
ΔSOFA score at T1-T0	−0.018 ± 0.055	−0.044	*0.799*
ΔLactate at T1-T0	0.089 ± 0.064	0.185	*0.173*
Total SOFA at T0	0.106 ± 0.074	0.192	*0.157*
Total SOFA at T1	0.088 ± 0.088	0.135	*0.323*
Lactate level at T0	−0.015 ± 0.068	−0.031	*0.823*
Lactate level at T1	0.073 ± 0.069	0.142	*0.295*
ScvO2 at T0	−0.29 ± 0.231	−0.168	*0.215*
ScvO2 at T1	−0.522 ± 0.221	−0.305	*0.022*

SE: standard error, P(v-a)CO_2_: venous-to-arterial carbon dioxide partial pressure difference, SOFA: sequential organ failure assessment, T0: within 1 h after admission to intensive care unit, T1: within 24 h after admission to intensive care unit.

**Table 6 biomedicines-13-02295-t006:** Predictors of postoperative 28 days mortality in patients who underwent surgery by univariate and multivariate logistic regression analysis in patients with P(v-a)CO_2_ at T0 > 6 mmHg.

	Univariate Analysis		Multivariate Analysis	
Parameters	OR (95% CI)	*p*-value	OR (95% CI)	*p*-value
Total SOFA score at T0	1.385 (1.026–1.871)	*0.034*	1.02 (0.615–1.692)	*0.94*
Total SOFA score at T1	1.473 (1.08–2.008)	*0.014*	1.471 (0.936–2.313)	*0.094*
ΔBicarbonate at T1-T0	0.667 (0.466–0.955)	*0.027*		
Lactate level at T1	1.28 (1.007–1.625)	*0.044*	0.864 (0.566–1.318)	*0.498*
ΔLactate at T1-T0	2.158 (1.184–3.931)	*0.012*		
ScvO_2_ level at T1	0.823 (0.698–0.971)	*0.021*	0.806 (0.636–1.022)	*0.218*

APACHE II: acute physiology and chronic health evaluation II, BMI: body mass index, CI: confidence interval, OR: odds ratio, P(v-a)CO_2_: venous-to-arterial carbon dioxide partial pressure difference, ScvO_2_: central venous oxygen saturation, SOFA: sequential organ failure assessment, T0: within 1 h after admission to intensive care unit, T1: within 24 h after admission to intensive care unit.

**Table 7 biomedicines-13-02295-t007:** Predictors of postoperative mechanical ventilator care in patients who underwent surgery by univariate and multivariate logistic regression analysis in patients with P(v-a)CO_2_ at T0 > 6 mmHg.

	Univariate Analysis		Multivariate Analysis	
Parameters	OR (95% CI)	*p*-value	OR (95% CI)	*p*-value
ΔP(v-a)CO_2_ Increase/Decrease status	0.238 (0.060–0.946)	*0.041*	0.265 (0.049–1.43)	*0.123*
ΔSOFA score at T1-T0	1.621 (1.087–2.416)	*0.018*	1.778 (1.136–2.784)	*0.012*
ScvO_2_ level at T1	0.908 (0.839–0.982)	*0.015*	0.898 (0.819–0.985)	*0.022*

APACHE II: acute physiology and chronic health evaluation II, BMI: body mass index, CI: confidence interval, OR: odds ratio, P(v-a)CO_2_: venous-to-arterial carbon dioxide partial pressure difference, ScvO_2_: central venous oxygen saturation, SOFA: sequential organ failure assessment, T0: within 1 h after admission to intensive care unit, T1: within 24 h after admission to intensive care unit.

## Data Availability

The research data that support the findings of this study are available on request from the corresponding author. The research data are not publicly available due to privacy or ethical restrictions.

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
