# Peer review of "Prognostic Significance of Venous-to-Arterial CO_2_ Difference in Critically Ill Patients After Major Abdominal Surgery"

_biomedicines, 2025, doi:10.3390/biomedicines13092295_

Round 1

Reviewer 1 Report (Previous Reviewer 1)

Comments and Suggestions for Authors

I appreciate the authors' efforts to revise the manuscript and address several of the concerns raised in the initial review. The study addresses an important clinical question regarding the prognostic utility of ΔP(v-a)CO₂ in postoperative critically ill patients and provides meaningful observations related to microcirculatory status and ICU outcomes. The inclusion of post-hoc power analysis, expanded discussion on ScvO₂ limitations, and clarification of temporal limitations are commendable. However, some critical issues remain insufficiently addressed. Notably, the authors have not provided a comparative analysis between included and excluded patients to assess potential selection bias from the substantial patient exclusions. Additionally, the manuscript still lacks composite outcome analysis for low-frequency events such as mortality, and no subgroup analysis was performed to explore the potential effects of laparoscopic or robotic surgery (pneumoperitoneum) on P(v-a)CO₂. Finally, the manuscript would benefit significantly from thorough English language editing, as grammatical issues and redundant phrasing persist throughout the text. These points should be addressed before the manuscript can be considered for publication.

Author Response

Reviewer 2 Report (New Reviewer)

Comments and Suggestions for Authors

Dear Authors,

First of all, congratulations for your work and for the opportunity to collaborate on this project.

I have carefully reviewed the article ,, Venous-to-Arterial Carbon Dioxide Tension Difference as a Useful Marker of Postoperative Morbidities in Critically Ill 3 Patients After Major Abdominal Surgery,,

Thus, in the following I will outline my observations and recommendations regarding this manuscript.

Title

Although it is technical, it is very loaded, so I suggest you find a simpler and easier to follow and understand formulation, e.g.: Prognostic Value of Venous-to arterial CO₂ Difference in Critically Ill Patients After Major Abdominal Surgery,,

You need to find a shorter, easier to read and understand title.

Abstract

A short section in which to explain the context, the importance of P(v-a)CO2 in perfusion assessment and the gap in the current literature.

After this section, clearly describe the objectives.

In the results section, report a few key results (3-4) that are directly relevant to the conclusion, not all secondary outcomes. Highlight outcomes with clinical impact

In the methodology section, briefly describe the study type and major inclusion criteria.

The conclusion states with certainty the value of the marker, which may seem problematic given the tone, small study sample, and single-center study. The wording of these should be moderated, “may serve as a useful marker,”

You should also write the results more clearly: ,, The group with a 21 ΔP(v-a)CO2>0(n=10,22.2% vs. n=7,54.5%, p=0.033 and n=4,8.9% vs. n=4,36.4%, p=0.02, respectively) had significantly lower use of mechanical ventilation and continuous renal 22 replacement therapy (CRRT); with shorter durations of both interventions(p=0.011 and 24 p=0.016, respectively),,??

The abstract must be rewritten with great care, it is very important!

Introduction

P(v-a)CO2 is primarily an indicator of the adequacy of cardiac output (flow) relative to CO2 production, not a pure marker of microcirculation. When cardiac output is low, CO2 clearance decreases, thus venous PCO2 increases relative to arterial PCO2. This mechanism is well described and anchored in physiology (Fick's principle for CO2) ,, Mallat J, Lemyze M, Tronchon L, Vallet B, Thevenin D. Use of venous-to-arterial carbon dioxide tension difference to guide resuscitation therapy in septic shock. World J Crit Care Med. 2016 Feb 4;5(1):47-56. doi: 10.5492/wjccm.v5.i1.47. PMID: 26855893; PMCID: PMC4733455.,, or in ,, Gavelli F, Teboul JL, Monnet X. How can CO2-derived indices guide resuscitation in critically ill patients? J Thorac Dis 2019;11(Suppl 11):S1528-S1537. doi: 10.21037/jtd.2019.07.10,,, . It is known to be associated with mortality, organ dysfunction, post-operative (high-risk surgery)

The usual pathological threshold is ~ 6 mmHg. In many studies and reviews, a threshold above 6 mmHg is used as a sign of hypoperfusion or inadequate flow, with poor prognosis in septic shock/high-risk surgery (Robin E, Futier E, Pires O, Fleyfel M, Tavernier B, Lebuffe G, Vallet B. Central venous-to-arterial carbon dioxide difference as a prognostic tool in high-risk surgical patients. Crit Care. 2015 May 13;19(1):227. doi: 10.1186/s13054-015-0917-6. PMID: 25967737; PMCID: PMC4486687.) (Huette P, Ellouze O, Abou-Arab O, Guinot PG. Venous-to-arterial pCO2 difference in high-risk surgical patients. J Thorac Dis. 2019 Jul;11(Suppl 11):S1551-S1557. doi: 10.21037/jtd.2019.01.109. PMID: 31388460; PMCID: PMC6642915.)

ScvO2 supplementation with lactate is recommended to deconspire residual hypoflow. (.

(Mallat J, Lemyze M, Tronchon L, Vallet B, Thevenin D. Use of venous-to-arterial carbon dioxide tension difference to guide resuscitation therapy in septic shock. World J Crit Care Med. 2016 Feb 4;5(1):47-56. doi: 10.5492/wjccm.v5.i1.47. PMID: 26855893; PMCID: PMC4733455)

Also worth seeing is the review published by Ltaief Z, 2021 in Critical care.

This chapter also requires great attention, it is not anchored in the literature, there are many published studies, which are not found in your research. It is known that the gap matters, that it correlates with microcirculation markers in major abdominal surgery. You need to describe very well the context, the data from the literature, the physiopathology and then direct towards the objectives and motivation current study.

Method

To clarify the surgical pathology and anesthetic risk of these patients.

Were there elective or scheduled surgeries? Are there possible deficient preoperative preparation?

Is invasive arterial and venous monitoring a routine practice or do patients with unfavorable intraoperative evolution or with an increased risk benefit from this type of monitoring?

Were the analyses performed with the same device for a long period of time? Also, temperature and type of heparinization can influence PCO2. There is a difference between the declared objective and the analysis actually performed. Practical analysis the main comparison was Δ≤0 vs Δ>0 in all patients.

Patient selection method? Did you basically take patients who had these catheters installed, for some reason?

Clinically important data are missing: type of interventions, duration/anesthesia, intraoperative losses, transfusion, use of vasopressors, septic shock/sepsis, fluid balance, mechanical ventilation (ventilator settings).

Vasopressors/inotropes and transfusion are key variables, they can modify both ScvO2 and CO2 clearance!

There is an increased risk of selection bias due to the lack of ABGA/VBGA in most patients.

A major revision of the method is needed

Results

You should define a single primary outcome (e.g. mechanical ventilation) and report the rest as secondary. Also clearly present the n in each analysis.

Discussion

The message,, monitoring changes could improve prognosis,, should be reconsidered. It is too strong for a retrospective study, without sufficient statistical support.

From the discussion it appears that patients with severe pulmonary dysfunction were excluded. This aspect is not found in the material and method (then should mechanically ventilated patients be excluded??)

Also, the impact of vasopressors, transfusion, blood loss, duration of intervention, patient pathology, anesthesia is not discussed?

Study limitations

Factors that may influence these indicators, require reporting and adjustment

Conclusions

It refers strictly to patients catheterized with both lines, which are a minority, not the entire post-surgical population.

Changes in 24-hour P(v-a)CO2 have a modest inverse correlation with ScvO2 and may support risk stratification in patients monitored invasively after major abdominal surgery, but given the retrospective design, small sample size, and restrictive selection, these results are hypothesis-generating and require confirmation in prospective, multicenter, adequately powered studies.

Round 2

Reviewer 1 Report (Previous Reviewer 1)

Comments and Suggestions for Authors

The authors have carefully addressed all of my comments, and I now consider the manuscript suitable for acceptance

Author Response

We would like to express our sincere gratitude to the reviewer for the careful review and the insightful comments. The thoughtful feedback has been invaluable in improving the quality and clarity of our manuscript.

Reviewer 2 Report (New Reviewer)

Comments and Suggestions for Authors

I congratulate the authors for the careful review of the manuscript:

"Prognostic Significance of Venous-to-Arterial CO₂ difference in 2 Critically Ill Patients After Major Abdominal Surgery,,. The previous observations have been largely integrated: the title is concise, the abstract restructured with clear objectives and moderate conclusions, the introduction is well anchored in the literature, and the methodology is described in detail, including important clinical factors. The results are rigorously presented, and the discussions and conclusions are appropriately nuanced. The only major problem that persists in my opinion is patient selection: out of 1275 initial patients, only 86 were analyzed, which implies a significant risk of selection bias and survivorship bias. The authors acknowledge this aspect in the limitations of the study, but I believe that the impact on the generalizability of the results should be emphasized more firmly in the discussion and conclusions section.

Author Response

This manuscript is a resubmission of an earlier submission. The following is a list of the peer review reports and author responses from that submission.

Round 1

Reviewer 1 Report

Comments and Suggestions for Authors

This manuscript explores the association between changes in P(v-a)CO₂ and postoperative complications in critically ill patients after major abdominal surgery. While the topic is clinically relevant and the study offers potential implications for early ICU monitoring and management, there are notable methodological and statistical limitations that require clarification and additional work before publication.

  1. The initial study population of 1,275 patients was reduced to 86 due to missing ABGA/VBGA data. This substantial reduction raises the risk of selection bias, as included patients may represent a pre-selected, more closely monitored subset. Include a detailed flowchart of excluded cases (beyond Figure 1) specifying exact exclusion reasons. Provide a comparison (if possible) between included and excluded patients to assess potential bias.
  2. The retrospective nature of the study and the small sample size—especially within the subgroup analysis (ΔP(v-a)CO₂>0, n=11)—limit the statistical power and generalizability of the findings. Acknowledge this more explicitly in the limitations section and consider performing a post-hoc power analysis for primary outcomes such as AKI, CRRT use, and ventilation duration.
  3. The study evaluates ΔP(v-a)CO₂ over a 24-hour period, but it is unclear how changes correlate temporally with clinical deterioration or intervention (e.g., was P(v-a)CO₂ reduction a cause or effect of stabilization?). Clarify whether changes in P(v-a)CO₂ preceded interventions like mechanical ventilation or CRRT initiation. If timing cannot be confirmed, clearly state this limitation.
  4. ScvO₂ is used as a surrogate for tissue perfusion, yet it remains a global indicator of oxygen balance and may not accurately reflect regional microcirculation, especially in septic patients with shunting or mitochondrial dysfunction. Tone down claims regarding ScvO₂ as a direct microcirculation marker. Include references acknowledging its limitations in this context.
  5. Some p-values suggest trends without clear clinical implications (e.g., 7-day mortality difference with 1 death in the ΔP(v-a)CO₂>0 group, p=0.041). The sample size is likely underpowered for rare outcomes. Interpret low-frequency outcomes with caution. Consider collapsing rare events (e.g., 7-day + 28-day + in-hospital mortality) or using composite endpoints if justifiable.
  6. Expand Discussion of Pneumoperitoneum: Laparoscopic/robotic approaches can elevate PaCO₂ intraoperatively. While the authors argue this is negligible after 24h, some residual effects may persist and should be acknowledged with supporting data or citations.
  7. Justify P(v-a)CO₂ Cutoff: The use of 6 mmHg as a threshold is based on prior studies, but local validation in your population (e.g., ROC analysis) would strengthen the argument.
  8. Clarify Use of SOFA/APACHE: The timing and interpretation of APACHE II and SOFA scores are sound, but consider presenting trends over the first 48–72h, if data are available.
  9. The manuscript would benefit from professional English editing. There are frequent grammar errors, awkward constructions, and redundancy in phrasing.

Reviewer 2 Report

Comments and Suggestions for Authors

The title should be changed into: ... ill patients after major abdominal surgery. And alos- should laparascopic surgery be listed as major abdominal surgery?

Introduction

  • third row- please exchange bleeding for hypoperfusion
  • Please explain stroke volume?
  • Is the blood viscosity really increased, often dilution due to excessive hydration can be present.
  • please exchange pressure targets into values
  • The last sentance in the first paragraph is unclear. Also, is tissue or endothelial injury directly caused by surgery?

M&M

  • why did you exclude patients who died within 48 hours- this pts were critically ill, however some of them lived long enough to acquire relevant data.
  • were the pts equiped with central venous catheter or they had it implanted?
  • Jugular vein was visualized and not confirmed by US
  • Interval between VBGA and ABGA was kept under 5 minutes- how were you able to obtain this data in retrospective study?- Please explain.
  • Intra-abdominal fluid collection- what did you mean by 3 cm in diameter?
  • Pleural effusion- FC sinus blurring was present on x-ray in which patient position? Majority of critically ill patients have X-ray examination in bed, so FC sinuses are often not blurred, there is a different pattern of fluid accumulation

Results

  • Included amount of patients is low (out of more than 1 200 pts in ICU)- the majority of pts was not critically ill and they did not need ABGA and VBGA monitoring?

Discussion

  • as I understood from the study, lactate levels correlated quite well with P (v-a) CO2 differences. Lactate is simple to obtain and is an established method for post-op patient evaluation. What would be additional benefit of a new maker?
  • Please explain on what is based your claim that clinical aplication of ScvO2 would be more beneficial in monitoring the condition of postoperative patients....
  • Is the tissue perfusion with new marker really more reliable as current markers?
  • In a previous study authors claimed that P (v-a)CO2 od 8.6 mm Hg was a treshold, in a current study 6 mmHg is a limit- Please explain.
  • Acknowledgements- perhaps authors should give it to all the hospital staff, taking care of patients and making this study feasible.